# Efficacy of the FIBROWALK Multicomponent Program Moved to a Virtual Setting for Patients with Fibromyalgia during the COVID-19 Pandemic: A Proof-of-Concept RCT Performed Alongside the State of Alarm in Spain

**DOI:** 10.3390/ijerph181910300

**Published:** 2021-09-30

**Authors:** Mayte Serrat, Mireia Coll-Omaña, Klara Albajes, Sílvia Solé, Miriam Almirall, Juan V. Luciano, Albert Feliu-Soler

**Affiliations:** 1Unitat d’Expertesa en Síndromes de Sensibilització Central, Servei de Reumatologia, Vall d’Hebron Hospital Universitari, Vall d’Hebron Barcelona Hospital Campus, 08035 Barcelona, Spain; malmirall@vhebron.net; 2Escoles Universitàries Gimbernat, Autonomous University of Barcelona, Bellaterra (Cerdanyola del Vallès), 08193 Barcelona, Spain; 3Eodyne Systems, Methodology, Methods, Models and Outcomes of Health and Social Sciences (M3O) Research Group, University of Vic, 08500 Vic, Spain; mireiaco4@blanquerna.url.edu; 4Department of Basics, Developmental and Educational Psychology, Faculty of Psychology, Autonomous University of Barcelona, Bellaterra (Cerdanyola del Vallès), 08193 Barcelona, Spain; k.albajes.eizaguirre@gmail.com; 5Faculty of Nursing and Physiotherapy, University of Lleida, 25198 Lleida, Spain; silvia.sole@udl.cat; 6Psychological Research in Fibromyalgia and Chronic Pain (AGORA Research Group), Institut de Recerca Sant Joan de Déu, Esplugues de Llobregat, 08950 Barcelona, Spain; JuanVicente.Luciano@uab.cat (J.V.L.); albert.feliu@uab.cat (A.F.-S.); 7Department of Clinical and Health Psychology, Faculty of Psychology, Autonomous University of Barcelona, Bellaterra (Cerdanyola del Vallès), 08193 Barcelona, Spain

**Keywords:** fibromyalgia, multicomponent treatment, pain neuroscience education, therapeutic exercise, cognitive behavioral therapy, mindfulness, randomized controlled trial, COVID-19, online treatment, teletherapy

## Abstract

FIBROWALK is a multicomponent program including pain neuroscience education, therapeutic exercise, cognitive behavioral therapy and mindfulness training that has recently been found to be effective in patients with fibromyalgia (FM). This RCT started before the COVID-19 pandemic and was moved to a virtual format (i.e., online videos) when the lockdown was declared in Spain. This study is aimed to evaluate the efficacy of a virtual FIBROWALK compared to Treatment-As-Usual (TAU) in patients with FM during the first state of alarm in Spain. A total of 151 patients with FM were randomized into two study arms: FIBROWALK plus TAU vs. TAU alone. The primary outcome was functional impairment. Secondary outcomes were kinesiophobia, anxiety and depressive symptomatology, and physical functioning. Differences between groups at post-treatment assessment were analyzed using Intention-To-Treat (ITT) and completer approaches. Baseline differences between clinical responders and non-responders were also explored. Statistically significant improvements with small-to-moderate effect sizes were observed in FIBROWALK+TAU vs. TAU regarding functional impairment and most secondary outcomes. In our study, the NNT was 5, which was, albeit modestly, indicative of an efficacious intervention. The results of this proof-of-concept RCT preliminarily support the efficacy of virtual FIBROWALK in patients with FM during the Spanish COVID-19 lockdown.

## 1. Introduction

Fibromyalgia (FM) is a chronic syndrome, of unknown etiology, characterized by widespread musculoskeletal pain and multiple concomitant symptoms, including fatigue and sleep disturbances, with an estimated prevalence of between 0.2 and 6.6% worldwide [1], and of 2.45% in Spain [2].

FM frequently coexists with psychological distress, anxiety, depression and other comorbid conditions such as chronic fatigue syndrome [1,3,4]. Although the etiopathogenesis of FM is still not clearly understood, a hypersensitization of the central nervous system characterized by an unbalanced function between descending inhibitory and facilitation pathways—which would facilitate hyperalgesia and allodynia—has been postulated [5,6]. The function of the descending nociceptive inhibitory pathway [6,7] is known to be altered by cognitive biases—such as maladaptive thoughts—along with emotional and behavioral factors, which, in turn, further potentiate the pain experience [8,9].

Current therapeutic strategies in FM usually combine pharmacological and non-pharmacological approaches [10,11,12,13], and multicomponent non-pharmacological treatments are currently considered the gold standard [12,14,15,16,17]. Regarding therapy components proved to be effective in FM, Pain Neuroscience Education (PNE) [18,19,20,21,22,23,24,25,26,27,28] is aimed at changing patients’ pain beliefs, emphasizing how overprotective behaviors can modulate pain experience [29,30,31], and it has been found to be effective for reducing pain disability, catastrophizing, avoidance behaviors and physical inactivity in patients with FM [32]. On the other hand, Cognitive Behavioral Therapy (CBT) and therapeutic exercise are also core pillars of intervention in FM [33,34,35], and combining both has been seen to be particularly effective at treating several FM symptoms [36,37,38] (e.g., relieving pain, fatigue, depression, and improving psychological well-being and physical functioning [12,33]). Furthermore, mindfulness training [39] has also been found to be an efficacious and cost-effective treatment for improving functional impairment, anxiety, depression, and quality of life in patients with FM [13,37]. The FIBROWALK program is an evidence-based multicomponent strategy based on the combination of the aforementioned therapeutic components with positive results in two recent RCTs in FM [40,41].

The outbreak of the coronavirus SARS-CoV-2 pandemic, associated mobility restrictions and the overloading of hospital services drastically changed the therapeutic paradigm in healthcare worldwide, with most non-urgent face-to-face therapies being canceled or postponed and replaced by telematic and virtual therapies. Patients with FM, similar to others with chronic medical conditions, were profoundly affected by these restrictions. In this regard, many studies indicate that the delay or the discontinuation of treatments in patients with chronic pain results in a deterioration of symptoms, a perception of pain aggravation and worsened quality of life [42,43,44].

The FIBROWALK program has proved to be an effective treatment for FM patients but, up to now, was only validated in hospital [41] and outdoor settings [40]. Given the need for moving effective treatments to the emerging field of telemedicine to continue providing support to patients with FM during the COVID-19 lockdown, the FIBROWALK program was adapted into a home-based virtual-format therapy. This adaptation was in line with the opinion of different international chronic pain expert panels at the beginning of pandemics (e.g., [45,46]), who prompted to rapidly implement telehealth support for patients with chronic pain to limit the impact of the pandemic on this particularly vulnerable population. Similar virtual approaches (e.g., using videos) have also been used by other health centers in Spain in the search of the benefit of patients with FM under quarantine. In this regard, Hernando-Garijo and colleagues [47] made available a home-based intervention of aerobic exercise during lockdown, finding, in a small sample size, RCT- positive results in pain intensity, functional impairment, pain catastrophizing and emotional distress compared to a passive control group. To date, as far as we know, no study has evaluated the efficacy of a virtual multicomponent treatment for FM in general or during the pandemic outbreak specifically.

Taking this state of the question as its foundation, this proof-of-concept RCT was aimed at making a preliminary assessment of the efficacy of the FIBROWALK multicomponent treatment, moved to a virtual format, as an add-on to Treatment-As-Usual (TAU). The efficacy was evaluated on functional impairment (as main outcome) and other secondary outcomes, such as kinesiophobia, anxiety and depression symptoms, and physical functioning. Specifically, we expected statistically significant and clinically relevant improvement in functional impairment and other secondary outcomes for FIBROWALK+TAU vs. TAU alone after treatment. Baseline differences between FIBROWALK responders vs. non-responders were explored to delimit potential patient characteristics related to the efficacy of the program, and the Number-Need-to-Treat (NNT) was calculated.

## 2. Materials and Methods

### 2.1. Design

A single-blind Randomized Controlled Trial (RCT) was conducted. Data were collected at baseline and at the end of the 12-week intervention. All procedures were conducted in accordance with the ethical standards laid down in the 1964 Declaration of Helsinki and subsequent updates. The FIBROWALK study protocol was approved by the Ethical Committee of Clinical Investigation of Vall d’Hebron Hospital (code: PR(AG)249/2020) and was registered at ClinicalTrials.gov NCT04284566. This study is reported according to the guidelines issued by the Consolidated Standards of Reporting Trials (CONSORT) [48].

### 2.2. Participants

A total of 151 participants diagnosed with FM, who met the selection criteria, and at the same time had been visited by the physical therapist from the Central Sensitivity Syndromes Specialized Unit (CSSSU) at the UHVH, were recruited from November 2019 to March 2020 to participate in this study. The treatment program was delivered from March 2020 to June 2020. The state of alarm was declared by the Spanish government on 14 March 2020 and lasted until 10 May 2020. This exceptional measure confined Spaniards in their homes. Inclusion criteria for the participants were: (a) FM diagnosis according to the 2010/2011 American College of Rheumatology (ACR) criteria, (b) ≥18 years old, (c) able to understand Spanish, and (d) able to provide written informed consent. Exclusion criteria were having terminal illnesses or programmed treatments that might interrupt the participation in the study.

### 2.3. Procedure

Participants were firstly recruited through an initial screening interview performed by the main researcher (MS), who provided an overview of the study. Informed written consent was obtained from all participants before the baseline examination. Participants were informed of their right to withdraw from the study at any time, with the guarantee that, if they desired, they could continue receiving their usual treatment. After completing baseline examinations, individuals who agreed to participate in the study were assigned to an alphanumeric code list and were randomized using the SPSS statistical package (v25) (Kaysville, UT, USA) to either the active group or control group (TAU). This process was carried out using numbered envelopes containing sheets with information regarding participant allocation. The envelopes were prepared by a nurse from the CSSSU. Neither the participants nor the clinical professional who conducted the program (MS) could be blinded. However, MS did not participate in any stage of the patient assessment process, and the researcher responsible for the outcome measures (MM) was blinded to the treatment allocation. All patients were evaluated before (baseline) and after (post) the 12-week treatment.

### 2.4. Treatment Arms

Participants allocated to the FIBROWALK arm underwent a multicomponent strategy based on therapeutic exercise, pain neuroscience education, CBT and mindfulness training. After the first face-to-face session, the FIBROWALK program was moved to a virtual format due to the proclamation of the Spanish state of alarm. From then on, a link to a 60 min video (hosted on a private YouTube channel) was sent by email once a week for the following 11 weeks (total time of the FIBROWALK program: 12 weeks). Each video sent provided detailed guidelines explaining how to perform different home-based aerobic exercises (such as walking down the hallway at home); education in the neuroscience of pain (based on the book “Explain Pain” by David Butler, Lorimer Moseley and Arte Sunyata and the educational recommendations of the Pain in Motion team led by Jo Nijs); and CBT based on the analysis of basic psychological processes and aimed at decreasing anxiety and depressive symptoms, at reducing pain catastrophizing and at changing inadequate emotional regulation strategies. For greater detail on the FIBROWALK contents, see the study by Serrat et al. [41]. A brief questionnaire (with 5 to 10 items) aimed at verifying if participants were actually involved in FIBROWALK virtual treatment was sent by email every week. These brief questionnaires asked for actual participants’ adherence to the proposed homework exercises (i.e., mindfulness practice, breathing exercises, relaxation training, therapeutic physical training) and also allowed us to check if FIBROWALK videos were watched or not. For the latter point, very basic concepts precisely explained in each video were checked (e.g., “Please, provide a short example of a catastrophic thought”) to ensure that particular video was watched. These weekly checks were used for the early detection of potential adherence problems (e.g., not watching the videos, not doing the exercises) as well as potential dropouts. The therapist (MS) contacted (via SMS and/or telephone calls) patients who did not answer the questionnaire or reported any issue related to the program or the study (e.g., not being able to do the homework, watch the videos, answer the questionnaire, etc.) and provided solutions for enhancing adherence. If needed, patients unable to watch or respond to the questionnaire in a specific week could ask for an extension date (e.g., watching/responding in a 2–3-week timeframe). Participants could also contact the therapist (via email) at any time if any other problem related to the treatment or the study arose.

The treatment as usual (TAU) provided to the control group patients of this study was mainly based on the administration of drugs adjusted to the symptomatic profile of each patient (amitriptyline, duloxetine, pregabalin and/or tramadol at low doses), with complementary advice on aerobic exercise adapted to the patients’ physical abilities and basic health education on their pathology. Patients in the control group remained on a waiting list so that at the end of the RCT they could benefit from the FIBROWALK treatment.

The prescribed drugs were not modified in any participant during the twelve weeks of the study. The use of rescue 1 g paracetamol tablets (maximum 1 tablet/8 h) was only allowed in the case of acute worsening of symptoms.

### 2.5. Study Measures

#### 2.5.1. Socio-Demographic and Clinical Characteristics

An ad hoc questionnaire gathering socio-demographic and clinical information was used to collect the following patient data: age, gender, educational level, employment situation, living arrangement (alone/accompanied), civil status, body mass index, illness self-perceived start/duration (in years), incapacity certificate obtained (indicating level of incapacity) or requested, and chronic fatigue syndrome diagnosis (issued by a rheumatologist).

#### 2.5.2. Primary Outcome

The Revised Fibromyalgia Impact Questionnaire (FIQR) [49] was used to measure participants’ functional impairment related to FM during the last week. It consists of 21 items answered on a 0–10 numerical scale and is divided into three dimensions: physical dysfunction (score ranges from 0 to 30), overall impact (score ranges from 0 to 20), and intensity of symptoms (scores from 0 to 50), with higher scores indicating greater functional impairment of FM. The FIQR might be considered the gold standard instrument for assessing multidimensional function/health-related quality of life (FIQR total score) and has been used in several RCTs as a main outcome for assessing the clinical impact of non-pharmacological interventions (e.g., 13, 40, 41). The Spanish version shows excellent internal consistency (Cronbach’s α = 0.91) [50]; in our sample, the α was 0.94.

#### 2.5.3. Secondary Outcomes

The Tampa Scale for Kinesiophobia (TSK) [51] was used to measure kinesiophobia. This scale has 11 items, which can be answered with a 4-point Likert scale (ranges from 0 to 11). Total scores of the TSK can go from 11 to 44 points, with higher scores indicating greater pain and fear of movement. The Spanish version shows adequate internal consistency (α = 0.79) [52]; in our sample, α = 0.90.

Depressive and anxiety symptomatology were measured with the Hospital Anxiety and Depression Scale (HADS) [53], which is divided into two dimensions (HADS-Anxiety and HADS-Depression), each dimension being further divided into 7 items, with a 4-point Likert scale response format. HADS-A and HADS-D total scores range from 0 to 21, with higher scores indicating higher symptom severity. The Spanish version shows adequate internal consistency for anxiety (α = 0.87) and for depression (α = 0.87) [54]; in our sample, α = 0.89.

The physical functioning component of the 36-Item Short Form Survey (SF-PF) [55] was used to measure physical functioning. This subscale comprises a total of 10 items, with a 3-point Likert scale response format. Total scores are transformed in order to range from 0 to 100, with higher scores indicating better physical functioning. The Spanish version shows adequate internal consistency (α = 0.94) [56]; in our sample, α = 0.86.

### 2.6. Statistical Analyses

Descriptive statistics were calculated for all variables and presented as means and standard deviations if continuous data were used, or absolute numbers and percentages (%) if categorical data were used. The Levene test was used to assess the equality of variances of continuous variables and Kolmogorov–Smirnov to test sample normality and distribution. Additionally, Student’s *t*-tests and χ^2^-tests were used to examine any possible baseline difference between FIBROWALK and the TAU group regarding sociodemographic and clinical characteristics.

We used analyses of covariance (ANCOVAs), including baseline values as a covariate, to examine differences between FIBROWALK+TAU vs. TAU alone at post-treatment in the FIQR scores and in each of the secondary outcomes. ANCOVA has been shown to have greater power to detect change than analysis of variance (ANOVA) in randomized study designs [57]. All outcomes were analyzed using the last observation carried forward (LOCF) method for imputing missing values in an Intention-To-Treat (ITT) approach. A sensitivity analysis was also conducted using the completer sample (i.e., completer approach). We also calculated the effect size (Cohen’s d) for each pairwise comparison, using the pooled baseline SD to measure the differences in the baseline–post mean values and to correct them for the population estimate [58]. The usual rule of thumb of d = 0.20, small; 0.50, moderate; and 0.80, large was used for interpreting observed effect sizes. In order to reduce the false discovery rate associated with multiple comparisons, we applied the Benjamini–Hochberg correction, which generates local significance levels where the first statistical test (the smallest *p*-value in ordered sequence) is assigned as the local significance level, which corresponds to the Bonferroni correction [59]. In addition, to assess the clinical significance of the improvement on the primary outcome (i.e., FIQR scores), we classified participants into two categories (i.e., responders vs. non-responders) according to their baseline-to-post change, using the criteria of ≥20% reduction in the pre–post FIQR total score [49]. This classification was then used to calculate the number needed to treat (NNT) in FIBROWALK+TAU compared to TAU. The NNT is an index aimed at helping clinicians to make results from RCTs more meaningful to them, and it refers to the estimated number of patients who need to be treated with a novel proposed treatment (here, FIBROWALK) for one additional patient to benefit (i.e., vs. the control arm). An NNT between 2 and 5 is considered to be indicative of a clinically effective treatment in pharmaceutical research [60]. A 95% CI for the NNT was also computed. T-tests and χ^2^-tests were also conducted to evaluate baseline differences between completers vs. non-completers (both groups), and between responders vs. non responders within the FIBROWALK+TAU arm.

All data analyses of the present study were performed using the SPSS v25 statistical software package.

## 3. Results

### 3.1. Participants’ Flow and Treatment Adherence

Of the 177 potential eligible participants, 26 were excluded at the screening for not meeting the selection criteria (i.e., not providing written informed consent). The initial study sample, consisting of 151 patients, was then randomized into the two study arms, with 75 and 76 individuals per arm (see Figure 1). All participants in the FIBROWALK+TAU group attended the 12 sessions of the program. The participants’ mean age was 54.35 years old (SD = 8.68; range: 22–76), the mean body mass index was indicative of overweight (M = 27.11, SD = 5.30), and the reported time since FM diagnosis was 15.75 years (SD = 9.16; range: 1–52). In the total sample, 20.5% of the patients were employed, 64.2% reported having a secondary education level or higher, and 90.1% also had a comorbid chronic fatigue syndrome diagnosis (more details in Table 1). Statistical differences between groups were found regarding the retention rate at post-treatment (FIBROWALK+TAU: 61.3%; TAU: 93.4%; χ^2^ = 22.27, *p* < 0.001). Non-completers from the FIBROWALK+TAU arm showed higher FIQR baseline scores than completers did (t = 2.047; *p* = 0.044). Non-completers from the TAU group showed a larger degree of incapacity than completers (χ^2^ = 9.78, *p* = 0.02) and lower anxiety levels than completers at baseline (t = −3.131; *p* = 0.007).

### 3.2. Baseline Differences between FIBROWALK+TAU and TAU Arms

As shown in Table 1, there were no statistically significant differences between groups in terms of demographic (except for civil status; *p* = 0.04) or baseline clinical characteristics.

### 3.3. Between-Group Differences in the Primary and Secondary Outcomes

All outcomes were analyzed following an ITT approach. Means and SD at baseline and post-test assessments (with imputed values) in both control and intervention groups are shown in Table 2. Significant treatment effects (*p* < 0.05) were found for functional impairment, anxiety, depressive symptoms and physical functioning (with effect sizes ranging between 0.23 and 0.48). A more robust treatment effect was found in the completer approach (Appendix A) where significant improvements were found for all outcomes (with d ranging from 0.33 and 0.89).

### 3.4. Baseline Differences between Responders and Non-Responders within the FIBROWALK+TAU Arm

As shown in Table 3, there were not significant differences between groups in terms of sociodemographic characteristics. However, there was a statistical difference in the physical functioning component of the SF-PF (*p* = 0.036), with non-responders presenting lower scores.

### 3.5. Number Needed to Treat (NNT)

Fourteen patients in the FIBROWALK+TAU group (30.43%; 14 out of 46) reached the status of responder at post-treatment (i.e., showed a decrease in their FIQR total score by at least 20% in comparison with their baseline assessment), whilst only seven patients in the TAU condition achieved this status (9.86%; 7 out of 71). The absolute risk reduction (ARR) in the FIBROWALK+TAU group in comparison with TAU was 20.58% (95% CI = 5.58 to 35.57%), with an NNT = 5 (95% CI = 2.8 to 17.9), meaning that five patients would need to be treated with FIBROWALK+TAU instead of TAU alone for one of them to become a responder.

## 4. Discussion

This RCT was aimed to preliminarily evaluate the efficacy of a virtual version of the FIBROWALK multicomponent program compared to Treatment-As-Usual (TAU) in patients with FM during the COVID-19 lockdown in Spain. As far as we know, it was the first study to evaluate the short-term efficacy of a virtual multicomponent treatment for the management of FM. The virtual FIBROWALK program arose as a necessity to provide health support for patients with FM when the COVID-19 pandemic broke out, and comprised weekly videos on PNE, home-based therapeutic exercise, CBT and mindfulness training.

### 4.1. Preliminary Efficacy of the Virtual FIBROWALK

The results of the present RCT show that the FIBROWALK intervention was efficacious (with small-to-moderate effect sizes) in reducing impairment, depression and anxiety symptoms and in improving physical functioning compared to TAU. Our results are consistent with previous research on the efficacy of online treatments for FM [61,62,63] and extend the already reported positive effects of FIBROWALK conducted in hospital [41] and outdoor [40] settings to virtual environments. It is worth noting that benefits associated with FIBROWALK administered in a virtual format were observed despite the extreme social circumstances happening when the RCT was carried out. Such circumstances were characterized by heavy restrictions of movement, fear, and tragically common traumatic events related to COVID infections and, especially, to the death of friends, relatives and acquittances. All these contingencies had a profound effect on the overall health of the population as a whole (e.g., [63,64,65,66,67]) and, particularly, among people with long-term conditions such as chronic pain and FM (e.g., [43,44,68]).

### 4.2. Components of the FIBROWALK

Multicomponent treatments involving physical exercise and cognitive behavioral strategies have shown to be some of the most effective approaches at improving physical function, psychological well-being and mental health status in patients with FM; and they are increasingly being recommended for evidence-based clinical guidelines [14,17]. Regarding therapeutic exercise, recent meta-analyses have supported its effectiveness for improving a wide range of FM symptoms, including pain and depressive symptomatology, and for increasing well-being and health-related quality of life [33]. Home-based aerobic conditioning has also been found to induce physiological and psychological benefits in patients with FM, including improvements in pain [69]. Relatedly, in a recent small (i.e., 17 individuals per arm) clinical trial conducted in Spanish patients with FM also during the COVID-19 lockdown [47], an intensive 15-week (2 sessions/week) telerehabilitation program based on aerobic exercise showed positive effects on pain intensity, functional impairment and psychological distress compared to a passive control group. On the other hand, CBT has been proven to be also efficacious (with small-to-medium effect sizes) for improving pain, health-related quality of life, negative mood, disability and fatigue in patients with FM [34,35]. In this regard, CBT is also strongly recommended by FM clinical guidelines and even advocated as the first step of stepwise treatment approaches for FM [17]. So, both exercise and CBT components may explain part of the therapeutic effects observed after virtual FIBROWALK completion.

Beyond therapeutic exercise and CBT, an important strength of the FIBROWALK protocol is the addition of PNE and mindfulness training, which constitute, respectively, a significant change of perspective regarding the classical pain education and the CBT focus on changing problematic thoughts. In this regard, PNE is aimed at educating patients on the mechanisms behind chronic pain, highlighting that any credible evidence of danger or safety in body tissues can increase or decrease pain perception, respectively [22], and has been found to be effective in patients with FM [18,19,20,21,22,23,24,25,26,27,28]. It is noteworthy that PNE seems even more effective when it is combined with therapeutic exercise, gradual exposure techniques, and CBT [24,70], all of which are integrated elements in the FIBROWALK program. On the other hand, mindfulness training is aimed at modifying the relationship with one’s thoughts—to be more nonjudgmentally open to them, with acceptance—rather than at changing their content [71]. It is also worth noting that mindfulness training was introduced in the late 1970s in hospitals as a program (i.e., Mindfulness-Based Stress Reduction) for helping people with problems in stress management and chronic health conditions (including chronic pain) to find alternative and more healthy ways of relating with personal life challenges [71]. Since then, several studies have reported that mindfulness may be an effective approach for promoting better mental health and fostering wellbeing in heterogeneous clinical and non-clinical populations (e.g., [72,73,74]). Furthermore, there is growing evidence that mindfulness can be also effective for improving core symptoms of FM. [37,75]. The role of mindfulness on wellbeing and mental health has also been studied during the COVID-19 pandemic. In this regard, some studies pointed out that mindfulness, as a trait, would be related with more effective coping during lockdown [76], and, as online intervention, it would be associated with improved wellbeing, stress and anxiety in various samples [77,78].

### 4.3. The Flourishing of Teletherapy during Pandemic and beyond

Virtual treatment programs such as FIBROWALK might be promising alternatives to conventional treatments in times of pandemic and beyond when it comes to specific logistic barriers, such as timing, travel or access difficulties in rural areas, or wellbeing barriers, such as patients’ fatigue, among others. They can also help in decongesting health system services, which are overstretched worldwide as a result of the COVID-19 pandemic. Furthermore, regarding cost-effectiveness, virtual treatments may also be highly advantageous to exhausted national health systems as they may help in considerably reducing healthcare personnel costs [79]. The implications of effective teletherapy approaches for FM are thorough, particularly in terms of increasing the availability of specific evidence-based treatments for FM. For instance, since healthcare professional time per patient is limited, having an effective self-administered online program available would imply that a professional could substantially increase the number of patients being treated without affecting the quality of the therapy delivered or that a professional could maintain the number of attended patients but increase the appointment time.

The COVID-19 pandemic has enforced the need for new approaches to treatment. The positive results of this proof-of-concept RCT on the virtual FIBROWALK program contribute to pave the way for a change in the paradigm of treatment in FM, potentially helping to overcome barriers for delivering effective treatments before, during and beyond the pandemic. Further RCTs evaluating the long-term effects of fully virtual FIBROWALK out of lockdown are also warranted.

### 4.4. Limitations

Several limitations to this study need to be acknowledged. As a proof-of-concept RCT, it did not have an active control group or a long follow-up period. Given that this virtual modality of FIBROWALK had never been tested before, we preferred an experimental design, treating this cohort of patients as a test sample. Promising clinical effects have been achieved in this study that warrant a further study with a more robust clinical design (larger sample size, including an active control group, with a long-term follow-up). In addition, it is important to highlight that this clinical trial was carried out in a specialized unit of tertiary referral hospital in the context of clinical practice. Related to the latter, strict selection criteria could not be established due to pressures in daily clinical practice (i.e., most patients were admitted), and, specifically, patients that requested the incapacity certificate were also included in this RCT. Additionally, for ethical reasons, a follow-up assessment for the TAU group was not feasible under lockdown circumstances and, accordingly, a pre-post design was used. Long-term follow-up assessments are essential for evaluating the effectiveness of multicomponent treatments in the context of real-world clinical practices and should be evaluated in further RCTs including the virtual FIBROWALK program. Moreover, as a result of treatment characteristics, participants were not blinded. Future studies should also include a second active arm (e.g., pain education with therapeutic exercise) for controlling treatment dosage. Although all patients allocated to FIBROWALK were asked to respond to a brief questionnaire each week about each sessions’ contents, there is also no complete certainty that patients had seen all the videos. Additionally, we could not gather information about participants’ personal losses throughout the RCT, which we know probably happened, as a higher number of deaths were observed worldwide (and particularly in Spain) at that time of the pandemic; this probably affected the results. Furthermore, patients included in the study presented high impact and duration of disease inasmuch as they were recruited from a specialized unit of a tertiary referral hospital, so the sample cannot be representative of all the population with FM. Further research of this treatment in primary care settings as well as in non-pandemic contexts might contribute to addressing these challenges. The online adaptation of the program was carried out as a result of the unexpected situation caused by the COVID-19 outbreak; this sudden program format adaptation—along with the variable IT knowledge of participants—may have contributed to increasing the dropout rate, which was higher than in previous RCTs using face-to-face FIBROWALK (38.7% vs. 23% [41]). However, it is also known that attrition rates are highly variable (4% to 54%) in online interventions for patients with chronic pain [80], so our results fell within the expected range. Recommendations on assessment domains in FM are mainly based on self-reported measures (e.g., core domain set for fibromyalgia in Outcome Measures in Rheumatology Clinical Trials (OMERACT; [81]). Notwithstanding this, further studies assessing the effects of virtual FIBROWALK should also include complementary objective outcomes such as aerobic fitness and other measures assessed by the clinician. Limitations in design (e.g., not having an active control group, not having complementary objective functional measures) along with the relatively high attrition may have caused an impact on the results of this proof-of-concept trial. In this regard, studies with more robust study designs are needed. Further RCTs aimed at evaluating the effects of a fully virtual version of FIBROWALK will also evaluate levels of attrition and explore potential ways for increasing treatment adherence.

## 5. Conclusions

This is the first study to demonstrate the short-term efficacy of a virtual multicomponent treatment compared to usual care for the management of FM during the COVID-19 lockdown in Spain. This proof-of-concept RCT preliminarily supports the efficacy of a novel and highly accessible virtual FIBROWALK program in the approach to FM to be used during times of the COVID-19 pandemic and beyond. Further RCTs including active control groups with an equivalent treatment dosage and assessing the long-term efficacy of the virtual FIBROWALK are warranted.

## Figures and Tables

**Figure 1 ijerph-18-10300-f001:**
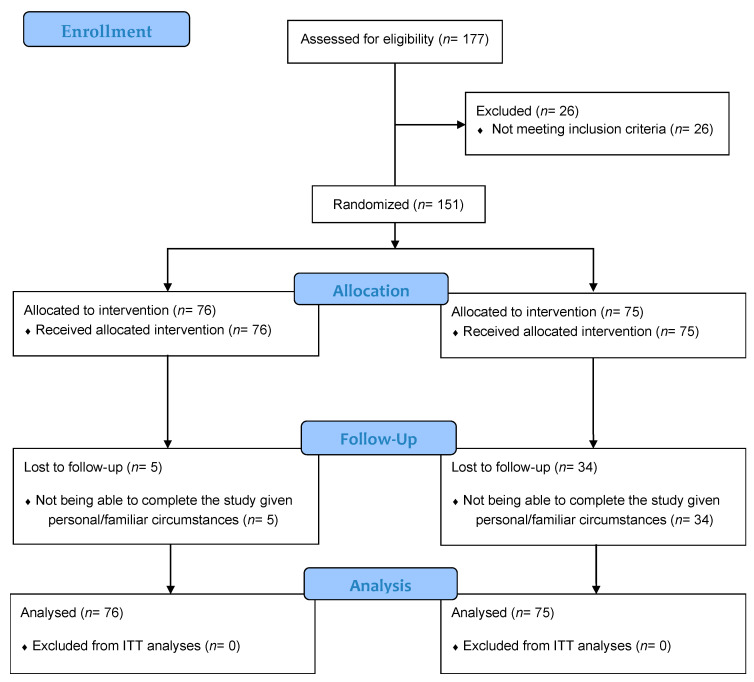
CONSORT Flow diagram of participants in the randomized controlled trial.

**Table 1 ijerph-18-10300-t001:** Demographic and baseline clinical characteristics by treatment group.

	FIBROWALK+TAU (*n* = 75)	TAU (*n* = 76)	t/χ²	*p*
**Age (years), M ± SD**	54.89 ± 8.94	53.82 ± 8.45	−0.761	0.448
**Women, *n* (%)**	71 (94.70)	70 (92.10)	0.400	0.745
**Civil Status, *n* (%)**			8.199	0.042
Single	13 (17.30)	15 (19.70)		
Married	39 (52.00)	52 (68.40)		
Divorced	19 (25.30)	7 (9.20)		
Widow	4 (5.30)	2 (2.60)		
**Not living Alone, *n* (%)**	65 (86.70)	65 (85.50)	0.041	0.840
**Educational Level, *n* (%)**			2.152	0.828
Without Studies	2 (2.70)	2 (2.60)		
Primary Education not completed	5 (6.70)	4 (5.30)		
Primary Education	17 (22.70)	23 (30.30)		
Secondary Education	32 (42.70)	29 (38.20)		
Higher Education	18 (24.00)	18 (23.70)		
Other	1 (1.30)	0 (0.00)		
**Employment Situation, *n* (%)**			11.249	0.128
Housekeeper	3 (4.00)	8 (10.50)		
Active	16 (21.30)	15 (19.70)		
On leave	12 (16.00)	15 (19.70)		
Unemployed with allowance	6 (8.00)	10 (32.20)		
Unemployed without allowance	8 (10.70)	1 (1.30)		
Retired	8 (10.70)	12 (15.80)		
Temporary work disability	8 (10.70)	6 (7.90)		
Other	14 (18.70)	9 (11.80)		
**Incapacity certificate, *n* (%)**			0.221	0.896
No	23 (30.70)	26 (34.20)		
Between 33% and 66%	47 (62.70)	45 (59.20)		
More than 66%	5 (6.70)	5 (6.60)		
**With incapacity certificate requested, *n* (%)**	24 (32)	30 (39.5)	0.918	0.397
**BMI, M ± SD**	27.39 ± 5.72	26.84 ± 4.89	−0.637	0.525
**ISPS, years, M ± SD**	16.97 ± 9.50	14.54 ± 8.71	−1.641	0.103
**With CFS, *n* (%)**	66 (88.00)	70 (92.10)	0.711	0.429
**FIQR, M ± SD**	71.83 ± 15.52	72.42 ± 15.93	0.230	0.818
**TSK, M ± SD**	27.93 ± 8.09	28.68 ± 8.10	0.570	0.569
**HADS Anxiety, M ± SD**	12.49 ± 5.00	12.71 ± 4.20	0.289	0.773
**HADS Depression, M ± SD**	11.80 ± 5.44	12.04 ± 4.65	0.291	0.771
**SF-PF, M ± SD**	27.00 ± 17.01	31.18 ± 19.32	1.412	0.160

Note: Statistically significant effects appear in **bold** (*p* ≤ 0.05). BMI: Body Mass Index; CFS: Chronic Fatigue Syndrome; FIQR: Revised Fibromyalgia Impact Questionnaire; HADS: Hospital Anxiety and Depression Scale; ISPS: Illness Self-Perceived Start; SF-PF: Physical Functioning component of the 36-Item Short Form Survey; TSK: Tampa Scale for Kinesiophobia.

**Table 2 ijerph-18-10300-t002:** Descriptive statistics and between-group analyses for primary and secondary outcomes from an ITT approach (with imputation of missing data).

	FIBROWALK+TAU (*n* = 75)		TAU (*n* = 76)		*F*	*p*	*d*
	Baseline	Post	Baseline	Post			
**Primary Outcome, M ± SD**							
**FIQR**	71.83 ± 15.52	66.31 ± 19.48	72.42 ± 15.93	72.66± 17.62	6.039	**0.015**	0.364
**Secondary Outcomes, M ± SD**							
**TSK**	27.93 ± 8.09	25.50 ± 11.69	28.68 ± 8.10	28.66 ± 7.46	3.841	0.052	0.296
**HADS Anxiety**	12.49 ± 5.00	11.75 ± 5.05	12.71 ± 4.20	13.04 ± 4.57	5.225	**0.024**	0.231
**HADS Depression**	11.80 ± 5.43	10.53 ± 5.82	12.04 ± 4.65	12.17 ± 4.85	6.483	**0.012**	0.276
**SF-PF**	27.16 ± 17.06	38.72 ± 22.91	31.18 ± 19.32	33.95± 19.55	9.349	**0.003**	0.480

Note: Statistically significant effects are shown in **bold** (*p* ≤ 0.05). When the Benjamini–Hochberg correction was applied to correct for multiple comparisons, all significant effects remained significant. FIQR: Revised Fibromyalgia Impact Questionnaire; HADS: Hospital Anxiety and Depression Scale; ISPS: Illness Self-Perceived Start; SF-PF: Physical Functioning component of the 36-Item Short Form Survey; TSK: Tampa Scale for Kinesiophobia.

**Table 3 ijerph-18-10300-t003:** Baseline differences between responders (FIQR ≥ 20%) and non-responders from the FIBROWALK+ TAU arm.

	Non-Responders (*n* = 32)	Responders (*n* = 14)	*t/*χ²	*p*
**Age (years), M ± SD**	54.63 ± 9.69	56.64 ± 8.33	−0.718	0.479
**Women, *n* (%)**	28 (87.50)	14 (100.00)	1.917	0.166
**Civil Status, *n* (%)**			6.783	0.079
Single	5 (15.60)	0 (0.00)		
Married	15 (46.90)	12 (85.70)		
Divorced	9 (28.10)	2 (14.30)		
Widow	3 (9.40)	0 (0.00)		
**Not living Alone, *n* (%)**	29 (90.60)	13 (92.90)	0.061	0.805
**Educational Level, *n* (%)**			2.864	0.721
Without Studies	1 (3.10)	0 (0.00)		
Primary Education not completed	1 (3.10)	2 (14.30)		
Primary Education	4 (12.50)	2 (14.30)		
Secondary Education	16 (50.00)	6 (42.90)		
Higher Education	9 (28.10)	4 (28.60)		
Other	1 (3.10)	0 (0.00)		
**Employment Situation, *n* (%)**			6.168	0.520
Housekeeper	0 (0.00)	2 (14.30)		
Active	9 (28.10)	3 (21.40)		
On leave	4 (12.50)	2 (14.30)		
Unemployed with allowance	1 (3.10)	1 (7.10)		
Unemployed without allowance	3 (9.40)	2 (14.30)		
Retired	4 (12.50)	1 (7.10)		
Temporary work disability	3 (9.40)	1 (7.10)		
Other	8 (25.00)	2 (4.30)		
**Incapacity certificate, *n* (%)**			3.753	0.153
No	10 (31.30)	8 (57.10)		
Between 33% and 66%	18 (56.30)	6 (42.90)		
More than 66%	4 (12.50)	0 (0.00)		
**With incapacity certificate requested, *n* (%)**	10 (31.30)	3 (21.40)	0.463	0.496
**BMI, M ± SD**	27.35 ± 5.66	25.59 ± 5.25	1.021	0.316
**ISPS, years, M ± SD**	14.69± 7.39	18.64 ± 12.41	−1.346	0.185
**With CFS, *n* (%)**	29 (90.60)	11 (78.60)	1.248	0.350
**FIQR, M ± SD**	70.74 ± 16.40	64.94 ± 14.59	1.139	0.261
**SF-PF, M ± SD**	25.78 ± 16.61	36.43 ± 11.84	−2.164	**0.036**
**TSK, M ± SD**	28.09 ± 7.91	28.00 ± 7.52	0.038	0.970
**HADS Anxiety, M ± SD**	13.19 ± 4.95	11.43 ± 4.70	1.125	0.267
**HADS Depression, M ± SD**	12.09 ± 5.29	10.07 ± 4.83	1.224	0.227

Note: Statistically significant effects appear in **bold** (*p* ≤ 0.05). BMI: Body Mass Index; CFS: Chronic Fatigue Syndrome; FIQR: Revised Fibromyalgia Impact Questionnaire; HADS: Hospital Anxiety and Depression Scale; ISPS: Illness Self-Perceived Start; SF-PF: Physical Functioning component of the 36-Item Short Form Survey; TSK: Tampa Scale for Kinesiophobia.

## Data Availability

The data that support the findings of this study are available from the corresponding author (M.S.) upon reasonable request.

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
