# Peer review of "Efficacy of the FIBROWALK Multicomponent Program Moved to a Virtual Setting for Patients with Fibromyalgia during the COVID-19 Pandemic: A Proof-of-Concept RCT Performed Alongside the State of Alarm in Spain"

_ijerph, 2021, doi:10.3390/ijerph181910300_

Round 1

Reviewer 1 Report

FIBROWALK is a multicomponent program including Pain Neuroscience education, therapeutic exercise, cognitive-behavioral therapy and mindfulness training, which was efficacious during Spanish lockdown. The outcome was functional impairment >.kinesio-phobia, anxiety and depressive symptomatology, self-reported physical function.  These results are of great theoretical significance and practical value for FIBROWALK multicomponent program. The analysis process is comprehensive, good organized, large amount of information and so on. Minor revision can be published in International Journal of Environmental Research and Public Health. However, there are some major issues need to be improved:

  1. Abstract: The readability needs to be further improved;
  2. Introduction: Some unnecessary references can be streamlined; Updated reference to supplement antidepressant and promote sleep of functional ingredientssuch as https://www.hindawi.com/journals/omcl/2018/3232080/
  3. Materials and Methods: In materials and methods the references involved in improved methods can be refined
  4. Results: Some variables use italics;Table 1and 3 are arranged on the same page as possible;
  5. Discussion:The discussion is rich, add attached titles to enhance readability, and highlight innovation points.
  6. Conclusions:The content is too little, so we need to be added;
  7. References: Authors shouldrevise the format of reference according to IJERPH Journal, In particular, some references lack pages.

Author Response

  1. Abstract: The readability needs to be further improved;

Authors: Thank you. We have modified the abstract for improving its readability.

  1. Introduction: Some unnecessary references can be streamlined; Updated reference to supplement antidepressant and promote sleep of functional ingredients such as https://www.hindawi.com/journals/omcl/2018/3232080/

Authors: We have added and cited a brief reference of the suggested article. See page 3,line 10, ref 10. In addition, following your suggestion, we have deleted some unnecessary references in this section old ref 21, 24 and 25.

  1. Materials and Methods: In materials and methods the references involved in improved methods can be refined

Authors: We have modified the description in the section 2.5.2 Primary Outcome in order to provide additional information.  See page 5, line 212-215.

The FIQR might be considered the gold standard instrument for assessing multidimensional function/Health-Related Quality Of Life (FIQR total score) and has been used in several RCT as a main outcome for assessing the clinical impact of non-pharmacological interventions (e.g. 13,40)”.

  1. Results: Some variables use italics; Table 1 and 3 are arranged on the same page as possible;

Authors: The use of italics throughout the manuscript has been standardized. Due to the number of files of Table 1 and 3, we could not arrange them into only one page and, at the same time.  However, we have reduced the space between them by following the format required by the journal, reducing the source of the tables to 8.

  1. Discussion: The discussion is rich, add attached titles to enhance readability, and highlight innovation points.

Authors: We have added attached tittles to the discussion as suggested. See from page 11, line 334.

4.1. Preliminary efficacy of the virtual FIBROWALK

4.2. Components of the FIBROWALK

4.3. The flourishing of teletherapy during pandemic and beyond

4.4. Limitations

  1. Conclusions: The content is too little, so we need to be added;

Authors: Thank you. We added some additional content in the conclusions section in the current version of the manuscript. See Page 13, line 457.

 “This is the first study to demonstrate the short-term efficacy of a virtual multicomponent treatment compared to usual care for the management of FM during the COVID-19 lockdown in Spain. This proof-of-concept RCT preliminarily supports the efficacy of a novel and highly accessible virtual FIBROWALK program in the approach to FM to be used during times of COVID-19 pandemic and beyond. Further RCTs including active control groups with equivalent treatment dosage and assessing long-term efficacy of the virtual FIBROWALK are warranted”.

  1. References: Authors should revise the format of reference according to IJERPH Journal, In particular, some references lack pages.

Authors: Adherence to journal format requirements has been checked and the manuscript accordingly adapted. We have revised all the references.

Reviewer 2 Report

Dear editor and authors, 

  There are a number of issues with this study.  

According to the authors patients with Fibromyalgia have dysfunctional illness beliefs but they do not back that up with any objective evidence. And the fact that other authors have stated the same doesn't make that true. This seems a clear example of opinion based medicine and it's important that the authors either remove it or state this.   

This study was registered with ClinicalTrials.gov Identifier: NCT04284566.

The study is described on that website in the following manner: "This is a two-arm RCT focused on the safety and potential efficacy of the multicomponent program FIBROWALK."  

Yet the article I was asked to review does not report on the safety of FIBROWALK despite the above.   

"The main objective of this RCT was...to improve functional impact." It has been known for a long time that subjective outcomes in exercise trials in ill people but also in healthy populations are very unreliable. However this study does not use objective outcomes. Even though it is also a non blinded study (by definition). Non blinded study that rely on subjective outcomes produce unreliable results.   

Moreover 45% (34/75) dropped out from the treatment group and of the people who followed the treatment, they do not know if they actually adhered to it.   

The study has a treatment as usual control group, in other words a no treatment group. This means that it did not have a proper control group.    The authors conclude that their treatment was effective, but if you take all the above into account then you have no idea if the small subjective improvement was down to the treatment under investigation or the design of the study. And the authors need to make that perfectly clear in their conclusion and abstract.   

On top of that, the authors conclude that the number needed to treat was 5. First of all that means that the treatment would only be effective in one out of every 5 patients treated. Or to put it differently, it means that it is not effective in 4 out of every 5 patients. That by definition is not an effective treatment. And that needs to be clearly stated in the abstract and conclusion. 

Author Response

According to the authors patients with Fibromyalgia have dysfunctional illness beliefs but they do not back that up with any objective evidence. And the fact that other authors have stated the same doesn't make that true. This seems a clear example of opinion based medicine and it's important that the authors either remove it or state this.   

Authors: Thank you. We agree with the reviewer. We deleted the sentence “maladaptive believes” which was not supported by any reference. Now the full sentence is “The function of the descending nociceptive inhibitory pathway [6,7] is known to be altered by cognitive biases -such as maladaptive thoughts-, along with emotional and behavioral factors, which, in turn, further potentiate the pain experience [8,9].” (Page 2, lines 68-70).

This study was registered with ClinicalTrials.gov Identifier: NCT04284566.

The study is described on that website in the following manner: "This is a two-arm RCT focused on the safety and potential efficacy of the multicomponent program FIBROWALK."  

Yet the article I was asked to review does not report on the safety of FIBROWALK despite the above. 

Authors: We thank you the reviewer for detecting this error in the Clinicaltrials website. We did not evaluate safety in this RCT, only efficacy. We will amend this in the protocol summary in ClinicalTrials.org.

"The main objective of this RCT was...to improve functional impact." It has been known for a long time that subjective outcomes in exercise trials in ill people but also in healthy populations are very unreliable. However this study does not use objective outcomes. Even though it is also a non blinded study (by definition). Non blinded study that rely on subjective outcomes produce unreliable results. 

Authors: Measures in this RCT (i.e., severity of FM, anxiety, depressive symptoms, kinesiophobia, and perceived physical function) are based on self-reported so blinding in assessment was not possible. Although, it is known that there exist little agreement among self-report, clinical examination and functional testing in chronic pain and also in general, and that is known that self-reported measures are subject to social desirability and recall biases, core outcomes in RCTs on FM are mainly based on self-report (e.g., functional status, health-realted quality of life, pain, fatigue...). In this regard, recommendations on assessment domains in FM are mainly based on self-reported measures (e.g., core domain set for fibromyalgia in Outcome Measures in Rheumatology Clinical Trials (OMERACT); Choi et al., 2009; Mease et al., 2009). Notwithstanding, further studies assessing the effects of virtual FIBROWALK should also include complementary objective outcomes such as aerobic fitness and other measures assessed by the clinician (e.g., WHODAS).

A large body of work in the pain area suggests that self-report measures are reliable and responsive to treatment effects (e.g. Rosier et al. 2002). The well-known variability in self-report is not necessarily error. There are no empirical studies to support self-report as a hindrance to treatment development and testing.

Choy, E. H., Arnold, L. M., Clauw, D. J., Crofford, L. J., Glass, J. M., Simon, L. S., ... & Mease, P. J. Content and criterion validity of the preliminary core dataset for clinical trials in fibromyalgia syndrome. The Journal of rheumatology 2009; 36(10), 2330-2334.

Mease, P., Arnold, L. M., Choy, E. H., Clauw, D. J., Crofford, L. J., Glass, J. M., ... & Williams, D. A. Fibromyalgia syndrome module at OMERACT 9: domain construct. The Journal of rheumatology 2009; 36(10), 2318-2329.

Moreover 45% (34/75) dropped out from the treatment group and of the people who followed the treatment, they do not know if they actually adhered to it. 

Authors: As stated in the discussion section, it is known that attrition rates can be high (up to 54%) in online interventions for patients with chronic pain (Buhrman, Gordh, & Andersson, 2016). However, it is also true that dropouts were higher than we expected in this RCT. Potentially, this increased dropout may be related with the exceptional circumstances during lockdown which potentially undermined patients’ predisposition to adhere to an online treatment. Furthermore, during these very first months of pandemic, many people experienced personal losses which may have a major impact on routines and compromises (such as completing a non-mandatory online program). We expect fewer dropouts in future studies assessing virtual FIBROWALK virtual out of lockdown. It is also worth mentioning that virtual FIBROWALK was rapidly designed for not totally discontinuing clinical support during lockdown, so further improved versions of the format may also have better associated adherence too.

Buhrman, M., Gordh, T., & Andersson, G. Internet interventions for chronic pain including headache: a systematic review. Internet Interventions 2016, 4, 17-34.

Regarding the last doubt raised by the reviewer (i.e., “do not know if they actually adhered to it “), brief online questionnaires were weekly administered to participants in order to verify that they actually watched the videos and did understand their content (lines 170 to 187 of the manuscript), so we could expect adequate adherence in those patients who finished the study. However, further refinements of the virtual program should also include automatized checks for an objective record of number of videos watched (i.e., attendance).

The study has a treatment as usual control group, in other words a no treatment group. This means that it did not have a proper control group. The authors conclude that their treatment was effective, but if you take all the above into account then you have no idea if the small subjective improvement was down to the treatment under investigation or the design of the study. And the authors need to make that perfectly clear in their conclusion and abstract.

Authors: The treatment as usual (TAU) provided to the control group patients of this study was mainly based on the administration of drugs adjusted to the symptomatic profile of each patient with complementary advice on aerobic exercise adapted to the patients' physical abilities and basic health education on their pathology. Following your comments, we underlined the preliminary nature of this RCT and the necessity to replicate these results in other RCTs including active control groups with similar treatment dosage.

See pag. 13, line 457 “This is the first study to demonstrate the short-term efficacy of a virtual multicomponent treatment compared to usual care for the management of FM during the COVID-19 lockdown in Spain. This proof-of-concept RCT preliminarily supports the efficacy of a novel and highly accessible virtual FIBROWALK program in the approach to FM to be used during times COVID-19 pandemic and beyond. Further RCTs including active control groups with equivalent treatment dosage and assessing long-term efficacy of the virtual FIBROWALK are warranted”.

See pag. 2 line 45 “The results of the proof-of-concept RCT preliminary supports the efficacy of virtual FIBROWALK was clinically efficacious in patients with FM during Spanish COVID-19 lockdown”.

On top of that, the authors conclude that the number needed to treat was 5. First of all that means that the treatment would only be effective in one out of every 5 patients treated. Or to put it differently, it means that it is not effective in 4 out of every 5 patients. That by definition is not an effective treatment. And that needs to be clearly stated in the abstract and conclusion. 

Authors: The NNT denotes the number of patients who must undergo an intervention to achieve 1 extra beneficial outcome; a NNT between 2-5 is considered indicative of the clinically effective treatment in pharmaceutical research (Cook & Sackett, 1998). Acknowledging this range for NNT, the virtual FIBROWALK administered during lockdown could be regarded as a clinical efficacious treatment, although that -in according to NNT- in a very modest way. Coherently with this interpretation, most of the effects in studied variables were found to be statistically significant but of small-to-moderate size. We tried to better delimitate with equipoise the modest impact of the treatment both at the abstract and conclusion sections.

Cook, R.J.; Sackett, D.L. The number needed to treat: a clinically useful measure of treatment effect. British Medical Journal, 310, 1998, 452-4.

Abstract Pag. 2 line 42 “In our study, the NNT was 5 which, albeit modestly, was indicative of an efficacious intervention”.

Discussion Pag.13 line 460 “This proof-of-concept RCT preliminary supports the efficacy of a novel and highly accessible virtual FIBROWALK program in the approach to FM to be used during times of COVID-19 pandemic and beyond. Further RCTs including active control groups with equivalent treatment dosage and assessing long-term efficacy of the virtual FIBROWALK are warranted”.

Reviewer 3 Report

This is an interesting proposal whose main objective was to evaluate the efficacy of a virtual FIBROWALK compared to treatment-as-usual (TAU) in patients with FM during the first state of alarm in Spain. The results obtained support that virtual FIBROWALK was clinically efficacious during Spanish lockdown, contributing to our knowlegde on the efficacy of virtual treatments. We need studies that address this issue not only due its ussefullness in these pandemic times, but also due to its potential practical applicability since technologies have an increasingly growing presence in our lives. Having said this, I have some concerns regarding the paper and I will elaborate on these concerns below.

I think the authors need to check and correct the format properly, in special in subheadings (e.g. line 119) and in the content of references section. Please, look carefully: it does not seem that the journal indicated format has been completely applied. 

- You should check if in this journal dois are mandatory if avalaible.

- You use two formats: left aligned and justified.

- You should check the use of tabs: sometimes there are in the first line of the first paragraph (e.g. line 57), sometimes not (e.g. line 188), and sometimes there are two (e.g. line 415).

- Line 114 maybe you must write vs. instead of vs

- Line 309 maybe you must delete this line.

- Line 330 please check tabulation, maybe you have an extra space to delete between line 329 and line 330.

- Line 387 maybe you must delete space after – (–to be more open instead of – to be more open).

- Line 396, maybe the order of reference could be [40, 81] instead of [81,40].

- Line 452 maybe you must write (38.7% vs. 23% [44]) instead of (38.7% vs. 23%; [44].

- Please check line 495: there are two points.

Perhaps it might be helpful to remember at the beginning of the discussion section in a more explicit way what the objective of the work was. While this is a fairly well-written study, it is also a complex contribution at some points. Please help the reader.

Finally, in my opinion, in view of the results and the quality of the discussion section, the conclusions section could be more ambitious.

Author Response

I think the authors need to check and correct the format properly, in special in subheadings (e.g. line 119) and in the content of references section. Please, look carefully: it does not seem that the journal indicated format has been completely applied. 

Authors: Thank you. It has been fixed in the current form of the manuscript. We have revised all the text, in special subheadings in according to the format of the journal.

- You should check if in this journal dois are mandatory if avalaible.

Authors: Thank you. The dois are not mandatory (if available) in IJERPH.

- You use two formats: left aligned and justified.

Authors: Fixed. All manuscript is now justified.

- You should check the use of tabs: sometimes there are in the first line of the first paragraph (e.g. line 57), sometimes not (e.g. line 188), and sometimes there are two (e.g. line 415).

Authors: Fixed. We have revised all the text to use the same tabulation format. Thank you.

- Line 114 maybe you must write vs. instead of vs

Authors: Fixed. Thank you.

- Line 309 maybe you must delete this line.

Authors: Fixed. Totally agree.

- Line 330 please check tabulation, maybe you have an extra space to delete between line 329 and line 330.

Authors: Fixed. Thank you.

- Line 387 maybe you must delete space after – (–to be more open instead of – to be more open).

Authors: Fixed. Thank you.

- Line 396, maybe the order of reference could be [40, 81] instead of [81,40].

Authors: Fixed. Thank you

- Line 452 maybe you must write (38.7% vs. 23% [44]) instead of (38.7% vs. 23%; [44].

Authors: Fixed. Thank you

- Please check line 495: there are two points.

Authors: Fixed. Thank you

Perhaps it might be helpful to remember at the beginning of the discussion section in a more explicit way what the objective of the work was. While this is a fairly well-written study, it is also a complex contribution at some points. Please help the reader.

Authors: Fixed. Thank you.

See page 11 lines 339-342 “This RCT was aimed to preliminary evaluate the efficacy of a virtual version of the FIBROWALK multicomponent program compared to treatment-as-usual (TAU) in patients with FM during the COVID-19 lockdown in Spain. As far as we know, it was the first study to evaluate the short-term efficacy of a virtual multicomponent treatment for the management of FM”.

Finally, in my opinion, in view of the results and the quality of the discussion section, the conclusions section could be more ambitious.

Authors: Thank you. We have added a sentence to give more emphasis to the results on page 11, lines 339-341

“This RCT was aimed to preliminarily evaluate the efficacy of a virtual version of the FIBROWALK multicomponent program compared to treatment-as-usual (TAU) in patients with FM during the COVID-19 lockdown in Spain. As far as we know, it was the first study to evaluate the short-term efficacy of a virtual multicomponent treatment for the management of FM.”

And on page 13, line 457.

“This is the first study to demonstrate the short-term efficacy of a virtual multicomponent treatment compared to usual care for the management of FM during the COVID-19 lockdown in Spain. This proof-of-concept RCT preliminarily supports the efficacy of a novel and highly accessible virtual FIBROWALK program in the approach to FM to be used during times of COVID-19 pandemic and beyond. Further RCTs including active control groups with equivalent treatment dosage and assessing long-term efficacy of the virtual FIBROWALK are warranted.”

Round 2

Reviewer 2 Report

Dea authors

You haven't addressed any of my comments and you continue to label a treatment effective yet a n n t or 5 indicates that it's not. Also you continue to ignore all the problems of your study design so you don't even know if the small subjective improvement in every 5th patient is caused by the treatment or your study design

Author Response

Response to Reviewers

#Reviewer 2

You haven't addressed any of my comments and you continue to label a treatment effective yet a NNT or 5 indicates that it's not. Also you continue to ignore all the problems of your study design so you don't even know if the small subjective improvement in every 5th patient is caused by the treatment or your study design.

Authors:

We sincerely thank you for taking the time to conduct this second review of the manuscript. Although the results in some study variables are modest in magnitude and there are some methodological shortcomings that temper the enthusiasm with our findings, we must disagree regarding your comment about the NNT.

We ask for your understanding that this RCT was conducted during hectic times (first strict lockdown in Spain), so there is not the chance to conduct this trial again and so neither to include some of the design improvements suggested by the reviewer (e.g., adding objective outcome variables, including an active control group, etc.). Given this fact, we think we can only acknowledge these study ‘flaws’ in the limitations section and label our RCT as “preliminary” or “proof-of-concept” trial. We believe that these limitations have been clearly and transparently shown in the previous version of the manuscript. Some examples below:

“The results of this proof-of-concept RCT preliminary supports the efficacy of virtual FIBROWALK in patients with FM during Spanish COVID-19 lockdown.” (Line 45, page 2)

“this proof-of-concept RCT was aimed at making a preliminary assessment of the efficacy of the FIBROWALK multicomponent treatment (Line 105, Page 3)”

“This RCT was aimed to preliminarily evaluate the efficacy of a virtual version of the FIBROWALK multicomponent program compared to treatment-as-usual (TAU) in patients with FM during the COVID-19 lockdown in Spain” (Line 339, Page 10).

“This proof-of-concept RCT preliminarily supports the efficacy of a novel and highly accessible virtual FIBROWALK program in the approach to FM to be used during times of COVID-19 pandemic and beyond. Further RCTs including active control groups with equivalent treatment dosage and assessing long-term efficacy of the virtual FIBROWALK are warranted.” (Line 477, Page 12)

Notwithstanding, following your suggestions, we have expanded the limitations section to include the idea that we cannot rule out that some of the observed improvements may also rely on design shortcomings: “Limitations in design (e.g., not having an active control group, not having complementary objective functional measures) along with the relatively high attrition rate may have caused an impact on the results of this proof-of-concept trial. In this regard, studies with more robust study designs are needed”. (Line 467, Page 12).

Moreover, we also acknowledged the limitations associated with self-report measures and included a new sentence in limitations section indicating that future studies should also include complementary objective outcomes as suggested by the reviewer:

Recommendations on assessment domains in FM are mainly based on self-reported measures (e.g., core domain set for fibromyalgia in Outcome Measures in Rheumatology Clinical Trials (OMERACT; [84]). Notwithstanding, further studies assessing the effects of virtual FIBROWALK should also include complementary objective outcomes such as aerobic fitness and other measures assessed by the clinician. (Line 462, Page 12).

However, it is worth bearing in mind that our trial used as a main outcome the Fibromyalgia Impact Questionnaire-Revised (FIQR), this instrument is currently considered as the gold-standard for RCTs in fibromyalgia (e.g., Luciano et al., 2012). Assessment protocol in this RCT is also aligned with Outcome Measures in Rheumatology Clinical Trials (OMERACT; Choi et al., 2009; Mease et al., 2009). Our group successfully published several RCTs evaluating the efficacy of different non-pharmacological approaches in fibromyalgia which used as primary outcome the FIQR:

Luciano, J. V., Guallar, J. A., Aguado, J., López-del-Hoyo, Y., Olivan, B., Magallón, R., ... & Garcia-Campayo, J. (2014). Effectiveness of group acceptance and commitment therapy for fibromyalgia: a 6-month randomized controlled trial (EFFIGACT study). PAIN, 155(4), 693-702.

Luciano, J. V., Martínez, N., Peñarrubia-María, M. T., Fernandez-Vergel, R., García-Campayo, J., Verduras, C., ... & Serrano-Blanco, A. (2011). Effectiveness of a psychoeducational treatment program implemented in general practice for fibromyalgia patients: a randomized controlled trial. The Clinical Journal of Pain, 27(5), 383-391.

Pérez-Aranda, A., Feliu-Soler, A., Montero-Marín, J., García-Campayo, J., Andrés-Rodríguez, L., Borràs, X., ... & Luciano, J. V. (2019). A randomized controlled efficacy trial of mindfulness-based stress reduction compared with an active control group and usual care for fibromyalgia: The EUDAIMON study. PAIN®, 160(11), 2508-2523.

Sanabria-Mazo, J. P., Montero-Marin, J., Feliu-Soler, A., Gasión, V., Navarro-Gil, M., Morillo-Sarto, H., ... & García-Campayo, J. (2020). Mindfulness-based program plus amygdala and insula retraining (MAIR) for the treatment of women with fibromyalgia: a pilot randomized controlled trial. Journal of Clinical Medicine, 9(10), 3246.

Serrat, M., Sanabria-Mazo, J. P., Almirall, M., Musté, M., Soler, A. F., Méndez-Ulrich, J. L., ... & Luciano, J. V. (2020). Effectiveness of a Multicomponent Treatment based on Pain Neuroscience Education, Therapeutic Exercise, Cognitive Behavioural Therapy, and Mindfulness in Patients with Fibromyalgia (FIBROWALK study): A Randomized Controlled Trial. Physical Therapy; pzab200. doi: 10.1093/ptj/pzab200.

The reviewer stated that we continue labeling “a treatment effective yet a NNT or 5 indicates that it's not “. As commented above, we agree that clinical effects of the multicomponent treatment were modest; however, we humbly think that stating that this number is indicative of lack of efficacy may be excessive. In this sense -as we informed in our previous response to reviewers- there is literature supporting that a NNT= 5 is indicative of an effective treatment (Cook & Sackett, 1998). In the current version of the manuscript, we acknowledge that a NNT of 5 could be indicative of a modestly effective intervention: “In our study, the NNT was 5 which, albeit modestly, was indicative of an efficacious intervention” (Line 44, Page 2).

Furthermore, to go into depth a little more on why we continue naming “modestly efficacious” a treatment with NNT= 5, it is worth to say that some authors suggest that there is not a “cut-off” for NNT at which an NNT must be considered ‘‘bad”. In this regard, NNTs should be judged differently based on the seriousness of the outcome (Shearer-Underhill & Marker, 2010), so any categoric classification of an intervention as “efficacious” OR “not efficacious” solely based on the NNT should be avoided. The cut-off point of ≥ 20% improvement in FIQR for detecting a clinically relevant change have been previously used in numerous studies (e.g., Pérez-Aranda et al., 2019; Sanabria-Mazo et al., 2020; Serrat et al., 2021), and it seems to be a reasonable cut-off even more when considering the hard social circumstances in which the trial was conducted. Additionally, we agree with Citrome (2007) when stating “…But if a difference in outcome is seen once in every 5 patients being treated with one intervention versus another (an NNT of 5), the result will likely influence day-to-day practice

To better delimitate the small-to-moderate effects of FIBROWALK, effect sizes were also calculated and reported for primary and secondary outcome measures:

“Statistically significant improvements with small-to-moderate effect sizes were observed in FIBROWALK+TAU vs. TAU regarding functional impairment and most secondary outcomes.” (Line 42, Page 2).

The results of the present RCT showed that the FIBROWALK intervention was efficacious (with small-to-moderate effect sizes) in reducing impairment, depression and anxiety symptoms and in improving physical functioning compared to TAU.” (Line 348, Page 10).

Other references in this response to reviewers:

Choy, E. H., Arnold, L. M., Clauw, D. J., Crofford, L. J., Glass, J. M., Simon, L. S., ... & Mease, P. J. Content and criterion validity of the preliminary core dataset for clinical trials in fibromyalgia syndrome. The Journal of rheumatology 2009; 36(10), 2330-2334.

Citrome, L. (2007). Dissecting clinical trials with'number needed to treat': calculation suggests a study's value to your patients. Current Psychiatry, 6(3), 66-72.

Luciano, J. V., Aguado, J., Serrano‐Blanco, A., Calandre, E. P., & Rodriguez‐Lopez, C. M. (2013). Dimensionality, reliability, and validity of the revised fibromyalgia impact questionnaire in two Spanish samples. Arthritis care & research, 65(10), 1682-1689.

Mease, P., Arnold, L. M., Choy, E. H., Clauw, D. J., Crofford, L. J., Glass, J. M., ... & Williams, D. A. Fibromyalgia syndrome module at OMERACT 9: domain construct. The Journal of rheumatology 2009; 36(10), 2318-2329.

Shearer-Underhill, C., & Marker, C. (2010). The use of the number needed to treat (NNT) in randomized clinical trials in psychological treatment. Clinical Psychology: Science and Practice, 17(1), 41–47.
